# Do Larger Language Models Generalize Better? A Scaling Law for Implicit Reasoning at Pretraining Time

## Abstract

Reasoning is an integral part of many tasks performed by language models (LMs). However, the effects of scaling model sizes and data on reasoning abilities at pretraining time remain understudied. To rigorously investigate this problem, we pretrain LMs from scratch on a synthetic implicit multihop reasoning environment designed to closely replicate the structure and distribution of real-world large-scale knowledge graphs. We then assess the LMs' ability to complete the missing edges in the graph, which requires multi-hop reasoning that can be viewed as a simplification of implicit reasoning during real-world pretraining. Interestingly, we observe that overparameterization can impair the implicit reasoning performance. We investigate different factors that affect the loss curve when scaling different components of the knowledge graph, model size, and training steps. To predict the optimal model size for a specific knowledge graph, we find an empirical scaling law that shows optimal-sized LMs can approximately reason over 0.008 bit information per parameter. This work shows counterintuitive effects of model size scaling and provides new insights into the relationship between scaling and reasoning in LLMs.

## 1 Introduction

Language Models (LMs) have demonstrated remarkable capabilities across a wide range of tasks, with reasoning being a core component (Wei et al., 2022a; Guo et al., 2025). While reasoning is typically enhanced during the post-training stage by encouraging LMs to generate long chain-of-thoughts (CoTs) (Guo et al., 2025; Yang et al., 2025), it is reasonable to assume that they already acquire the foundations of such capability during pretraining, given that post-training operates at a significantly smaller scale. Several recent studies have explored the mechanisms by which LMs may acquire reasoning-related abilities through next-token prediction pretraining (Zhu et al., 2024; Wang et al., 2024a;b). However, the impact of scaling on LMs' reasoning ability during pretraining remains poorly understood.

The general scaling behavior of LMs at pretraining time has been extensively investigated, including the well-known exponential scaling laws for testing loss and compute proposed by Kaplan et al. (2020) and the training compute-optimal scaling studied by Hoffmann et al. (2022a). Recent work has also examined the scaling of specific capabilities like machine translation (Ghorbani et al., 2022) and knowledge capacity/memorization (Allen-Zhu & Li, 2025; Lu et al., 2024). According to these existing scaling laws, it is in general believed that larger models imply better testing loss or task performance.

In this paper, instead we find that the scaling of LMs' reasoning capability at pretraining time differs from normal power-law scaling, in a simplified controlled pretraining environment. We use **implicit reasoning** to denote the reasoning behavior that naturally emerges during pretraining. i.e. *the capability to draw new conclusions from existing knowledge without being explicitly trained to generate chain-of-thoughts (CoTs).* More specifically, we define implicit reasoning over world knowledge as the task of completing missing edges in an incomplete knowledge graph, which requires multi-hop traversal according to predefined logic rules that are implicitly encoded in the graph generation process. To investigate this, we pretrain LMs from scratch using only triples from the incomplete graph and then evaluate their ability to infer the missing connections.

With sufficient compute, we find that the curve of implicit reasoning loss versus model size follows a U-shape, revealing an **optimal model size** that yields the best reasoning performance. This suggests that overparameterization may impair the implicit reasoning capability instilled during pretraining. We first observe this phenomenon using data derived from real-world knowledge graphs, and then systematically study it with synthetically generated data.

We investigate important factors that affect the U-shaped scaling of reasoning loss versus language model size. Our important findings can be summarized as follows:

- The minimum reasoning loss reachable by an LM is solely determined by the training data, regardless of training steps and model size.
- The optimal model size is solely determined by knowledge graph complexity and data size regardless of training steps.
- We show that an optimal-sized LM can approximately reason over 0.008 bit information per parameter.

As we observed that the **optimal model size** is likely solely determined by the training knowledge graph, we then aim to find an empirical scaling law that can predict the optimal model size from knowledge graph statistics. We identify a linear relationship between the optimal model size and our proposed **graph search entropy**, which quantifies the entropy of performing random searches on a knowledge graph. Under this framework, we find that each parameter in the optimal model size can reason over approximately 0.008 bits of information in a knowledge graph. In contrast, Allen-Zhu & Li (2025) show that a language model can memorize up to 2 bits of information per parameter—substantially more than its reasoning capacity. This gap arises both from the greater difficulty of reasoning compared to memorization and from the different methodologies used to compute these information quantities. A more detailed discussion is provided in Section 5.2.

Our work contributes to the broader understanding of LLM reasoning by shedding light on the intricate relationship between scaling and implicit reasoning capability. Our proposed empirical reasoning scaling law provides possible practical insights for optimizing LLMs' implicit reasoning ability at pretraining time.

## 2 METHOD

While real-world LLMs are pretrained on large scale text corpora, this corpus can be viewed as encoding a wide range of world knowledge. The power of LLMs lies in the fact that they can not only memorize the world knowledge and extract the knowledge when queried, but also reason over the world knowledge and draw novel conclusions. In this paper, we propose constructing a simplified pretraining corpus from a knowledge graph. A knowledge graph is comprised of a set of (head entity, relation, tail entity) triples, and we use each knowledge triple as a training example. We test the reasoning capability of a language model trained on such a corpus by testing its accuracy in completing triples that have never been seen in the knowledge graph but can be deduced through latent rules encoded in the graph structure. For example, if we know A is B's father, and B is C's father, then we can deduce that A is C's grandfather.

Formally, a knowledge graph $G$ consists of $|G| = N$ triples $(e^h, r, e^t)$, where $e^h \in \mathcal{E}$ is the head entity, $e^t \in \mathcal{E}$ is the tail entity, and $r \in \mathcal{R}$ is a relation. A simple example of knowledge triple is (DC, is the capital of, USA). These knowledge triples naturally form a graph, with nodes as the entities and each edge labeled with a relation type. We denote the total number of entities or nodes by $|\mathcal{E}| = N_e$ and the total number of edge or relation types by $|\mathcal{R}| = N_r$. Then a corpus constructed from this knowledge graph would consist of $N$ data points. The objective of a language model with the next token prediction loss with parameter $\theta$ trained on this corpus is then:

$$L(\theta) = \arg\min_{\theta} \frac{1}{N} \sum_{i=1}^{N} -\log P_\theta(e_i^h, r_i, e_i^t).$$

To eliminate confounding variables and information contained in the lexical form of the entity and relation names, we label each entity and relation with a random ID and tokenize the IDs by characters. We use the Llama (Touvron et al., 2023) model architecture to implement LMs of different sizes by

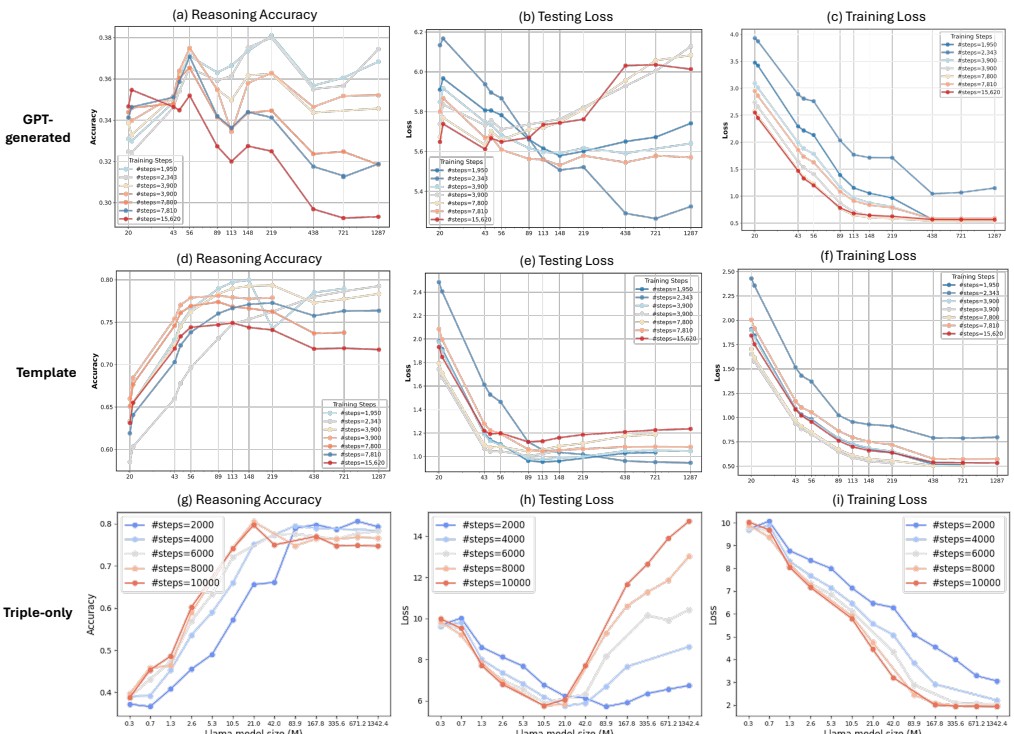

Figure 1: The multiple-choice accuracy/loss on unseen triples of different-sized LMs trained on a real-word knowledge graph FB15K-237. The first column shows that the testing accuracy decreases after a certain model size. The second column shows U-shape loss curves of LMs trained with different numbers of steps. The third column shows the training loss decreases steadily. These trends are stable across different ways of processing the knowledge triples, with the triple-only data shows the cleanest trend. Note that the model size on x-axis is in log scale.

adjusting the hidden dimensions and the number of layers. The specific parameter scheme can be found in the Appendix B.

To evaluate the language model's capability of reasoning over the knowledge graph, we test the LMs on a held-out set of triples that are not seen in the training time. Note that all entity and relation types should have been seen during training time and the language model is only tasked to connect missing edges. To eliminate the need to generate the correct form of relation and entity IDs, and to handle the case where multiple correct answers exist, we design the testing set to be 10-option multiple-choice questions: the language model is tasked to choose the correct tail entity given the head entity and the relation. We ensure that there is only one correct answer among the given 10 options. Suppose there are $M$ questions in the testing set.[1] For a ground truth triple $(e^h, r, e^t)$, we design 9 distracting options $e^{(1)}, e^{(2)}, ..., e^{(9)}$. Then we use the test accuracy $\text{Acc}(\theta, G)$ and testing loss $\ell(\theta, G)$ (the next token prediction loss) to evaluate the reasoning capability of a language model $\theta$ over the knowledge graph $G$:

$$\hat{e}_i = \arg \max_{e \in \{e_i^t, e_i^{(1)}, e_i^{(2)}, ..., e_i^{(9)}\}} P_\theta(e|e_i^h, r_i),$$

$$\text{Acc}(\theta, G) = \sum_{i=1}^{M} \mathbb{1}[\hat{e}_i = e_i^t]/M, \qquad \ell(\theta, G) = \sum_{i=1}^{M} -\log P_\theta(e_i^t|e_i^h, r_i)/M.$$

## 3 INITIAL EXPERIMENTS WITH REAL-WORLD KNOWLEDGE GRAPH

In our initial sets of experiments, we investigate the reasoning scaling effect using a real-world knowledge graph, FB15K-237 (Toutanova & Chen, 2015). FB15K-237 is sampled from FB15K

---

[1]We fix $M = 1000$ for all of our experiments.

(Bordes et al., 2013), which is a dataset adapted from the Freebase knowledge base (Bollacker et al., 2007), a web-scale knowledge base released by Google. FB15K-237 contains $N_e = 14,505$ entities, $N_r = 237$ relations, and $N = 310,116$ knowledge triples. We process this dataset in three different ways: (a) translate each knowledge triple into a natural language sentence by prompting GPT4 and then tokenize the sentence with a pre-trained tokenizer, as shown in the first row of Figure 1; (b) translate each knowledge triple into a natural language sentence using pre-generated templates, as show in the second row of Figure 1; (c) translate each knowledge triple into text by assigning a random ID to each entity and relation and tokenize them by characters, as shown in the last row of Figure 1. An example can be found in Appendix A Figure 5.

In Figure 1, we show different-sized LMs trained on FB15K-237 in all settings with different numbers of training steps. We observe a consistant reasoning performance drop when using larger models, across different ways of processing the knowledge triples, while the training loss decreases monotonically with respect to model size. This observation contradicts the previous belief that larger models always yield a smaller testing loss.

This implies that a language model can overfit to the training data when it is overparameterized for the underlying reasoning structure. Such deviation from traditional scaling law has also been reported in broken neural scaling law (Caballero et al., 2023) which proposed a double-descent-like (Nakkiran et al., 2020) function form instead of a monotonic power-law form. There have also been observations of tasks with inverse scaling (Wei et al., 2023) for large LMs.

In this paper, we focus primarily on the scaling of model size and data complexity. Rather than merely increasing the size of the training data, we explore many different setting for generating synthetic knowledge graphs. This allows us to ablate individual components of the graph generation process and examine how overall graph complexity affects reasoning. In the synthetic experiments presented below, we use random IDs instead of natural language sentences to eliminate lexical and syntactic effects, yielding cleaner trends from which we can draw quantitative conclusions.

In the following sections, we will mostly focus on understanding the "turning point" of the reasoning loss. More specifically, we want to understand what is the **optimal model size**, that is the model size that can obtain the smallest possible reasoning testing loss. As shown in Figure 1 and in Figure 3 (a), we find this optimal model size is largely stable when training the model for enough steps. Note that, at training time, we repeat the training triples for many epochs (e.g. 30 times for FB15K-237) to find the optimal model size. This graph epoch is different from the real-world cases where we repeat the whole pretraining corpus for certain epochs. Because we can view each triple in the graph as a piece of factual knowledge (e.g. Barack Obama's wife is Michelle Obama), this knowledge is usually repeated many times in a pretraining text corpus, in many different forms. Therefore, although our models have seen the same triple many times during training, the same piece of factual knowledge could also have been repeated several times in one pass of a real-world pretraining corpus.

## 4 SYNTHETIC DATA CONSTRUCTION

To investigate how the underlying knowledge structure influences LMs' reasoning performance, we propose an algorithm to generate synthetic knowledge graphs that mimic real-world knowledge graphs. More specifically, we assume that the knowledge graph generation process is governed by a set of logical rules.

For example, a rule for inferring the locatedIn relation can be $(e_1, \text{locatedIn}, e_2) \leftarrow (e_1, \text{neighborOf}, e_3) \land (e_3, \text{locatedIn}, e_2)$. Formally, for a target relation $r$, we consider logic rules with conjunctive form. For $\forall \{e_i\}_{i=0}^n \subset \mathcal{E}$,

$$(e_0, r, e_n) \leftarrow (e_0, r_1, e_1) \land ... \land (e_{n-1}, r_n, e_n),$$

where $(e_{i-1}, r_i, e_i) \in \mathcal{G}$. We abbreviate such rule by $h(r) = [r_1, r_2, ..., r_n]$. We randomly generate a set of logical rules $\mathcal{H}$ and ensure there are no cycles in the set. To grow a graph that follows these rules,

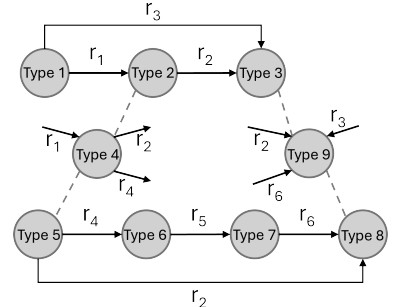

Figure 2: Nine possible node types generated by two logical rules. Each entity position in a rule would create a new entity type. Each relation shared between two rules would also create two new entity types.

we enforce sparsity of the possible relation types connecting to and branching out each entity. More specifically, we define *node types* based on the possible relation types connecting to and branching out each entity, based on the generated rules, as illustrated in Figure 2. Such sparsity is also observed in real-world knowledge graphs.

Our random graph generation process is inspired by the preferential attachment process (Barabási & Albert, 1999), which is used for generating scale-free networks with a power-law distribution for the degrees of the nodes. Intuitively, preferential attachment implies a "the rich get richer" approach to edge placement in the graph. Each time a new node is added to the graph, there is a 'preference' to connect to the nodes that are already highly connected, with a probability proportional to the target node's degree. Since we have observed the scale-free property in real-world knowledge graphs and the internet is known to be a scale-free network (Albert et al., 1999), we adopt a preferential attachment based graph generation process. To accommodate different relation types assigned to each edge, we maintain a degree distribution for each relationship and add new edges according to preferential attachment. A comparison of the node degree distribution between synthetic graph and real-world graph can be found in Appendix C Figure 6.

The code for our random graph generation algorithm is shown in the Appendix D. In summary, we first randomly generate a set of rules $\mathcal{H}$, with the number of rules $|\mathcal{H}| = N_h$ and the range of rule length $[L_{min}, L_{max}]$ as hyperparameters. Then we generate all possible node types as illustrated in Figure 2, with the maximum number of relations per node $M_r$ as a hyperparameter. We generate a seed graph by instantiating each rule with a set of new entities. To this, we incrementally add one new entity until the number of entities reaches $N_r$, by first randomly assigning a node type to it, and then randomly sampling the $m$ relation types from the set of relations defined by the node type. We choose the target of these $m$ new edges by preferential attachment. After adding every $K$ entities, we search through the current graph to add any edges that can be inferred through the logic rules defined in $\mathcal{H}$. We call the triples that can be deduced through a logic rule by *deducible triples*, otherwise *atomic triples*.

Finally, we limit the number of training triples to $N$ and ensure that the the ratio between the number of deductible triples and atomic triples to $\gamma$ by subsampling the generated graph. We also further ensure that the triples in the held-out test set are all deductible through the training triple. In this way, we can generate synthetic knowledge graphs with specific sizes and complexity.

## 5 SCALING LAWS

In this section, we investigate the scaling law of language models trained on different synthetic knowledge graphs. We conduct controlled experiments to show the effect of individual components of the data generation process. We also propose an information-theoretical way to measure the overall reasoning complexity of a knowledge graph, which we call the **graph search entropy**, and relate this linearly with the **optimal model size**. i.e. the model size that obtains the lowest possible testing loss.

### 5.1 GRAPH GENERATION ABLATION

We study the effects of the following four hyperparameters of graph data generation: the number of triples $N$, the number of entities $N_e$, the number of relations $N_r$, and the number of rules $N_h$. We fix all training hyperparameters as specified in the Appendix B. In all experiments except Figure 3 (a), we train all models for 10k steps. The detailed data generation configuration for each set of experiments can also be found in the Appendix B.

**Stable optimal model size with respect to training steps.** In Figure 3 (a), we show the effect of training language models on the same knowledge graph with different numbers of training steps. As mentioned in Section 3, the optimal model size becomes smaller when the number of training steps increases, and then becomes stable after 4k steps. Another observation is regardless of the number of training steps, the maximum accuracy or minimum loss is stable. While we have ensured that all testing triples can be deduced through the training triples, there seems to be a performance cap determined solely by the knowledge graph data, which is unaffected by model size. So in the following experiments, we choose to train models with a large number of training steps to ensure we capture the optimal model size.

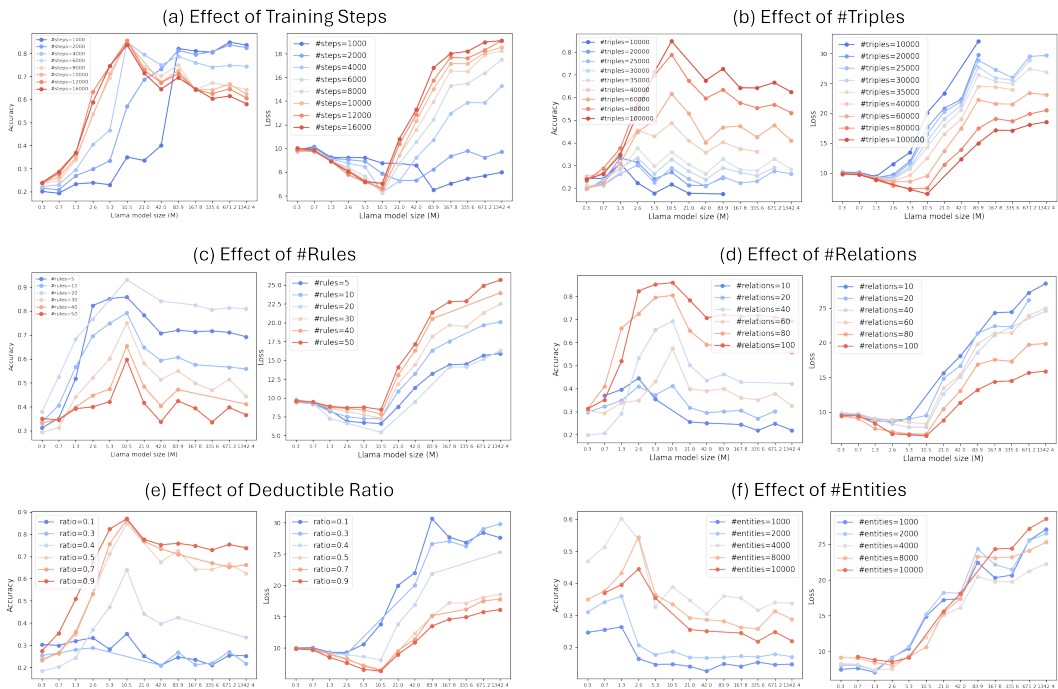

Figure 3: We show the effect of different hyperparameters of the synthetic knowledge graph generation process. In each experiment, we keep all other parameters the same and only change one hyperparameter. We show the effect with both the testing accuracy (left) and the testing loss (right) as the y-axis, with different model sizes as the x-axis in log scale.

**More triples implies a larger optimal model size.** In Figure 3 (b), we show the effect of the number of unique triples $N$ sampled after the same knowledge graph generation process. This setting is arguably the most similar to the real-world pretraining of language models: the underlying world knowledge graph of all the pretraining corpora is largely stable, and training data are realizations of the underlying knowledge graph and so the sizes of different corpora are simply a result of subsampling/upsampling the knowledge in the existing graph. We can see that a larger number of training triples results in a larger optimal model size and a better reasoning performance. This observation aligns with the classic scaling laws. However, there exists an optimal model size for the full knowledge graph: after sampling beyond the size of the full knowledge graph, you can only sample previously seen knowledge. In this case, the optimal model size would be stable no matter the training data size.

**Number of rules does not impact optimal model size.** In Figure 3 (c), we show the effect of generating knowledge graphs of the same size with different numbers of rules $N_h$. More rules mean that the testing triples need to be solved in more ways. The number of rules does not have a significant effect on the optimal model size, but affects the reasoning performance. There appears to be an optimal number of rules (20) that results in the best performance. This is because more rules increases the complexity of solving the test set while fewer rules increases the ambiguity in the training set. i.e. a relation may be be deduced through correlations outside of the predefined rules. The reason why the number of rules does not affect the optimal model size is likely because it does not significantly impact the graph search entropy. This will be discussed in detail in Section 5.2.

**More relations imply a larger optimal model size.** In Figure 3 (d), we show the effect of generating knowledge graphs of the same size and the same number of rules with different numbers of relations $N_r$. While the rules used for deducing the testing set remain the same for all experiments, there are additional relations that may not be used by any of the rules. We construct knowledge graphs with an excessive number of relations by adding additional relation patterns. In general, more relations improves the best reasoning performance while increasing the optimal model size. More relations increases the complexity of the knowledge graph, and thus increases the optimal model size. On the other hand, as discussed in the previous experiment, a small number of rules along with a small

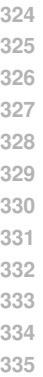
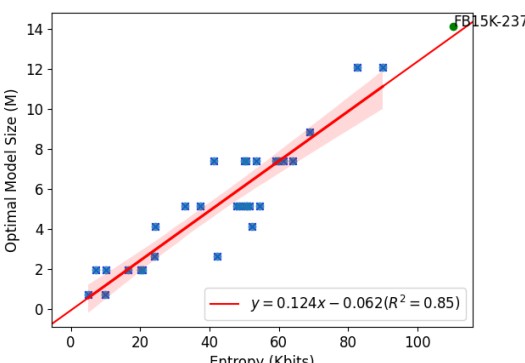

Figure 4: The optimal model size with the lowest possible testing loss v.s. the graph search entropy. The red line is the linear regression line using data from the synthetic experiments (blue squares), with a 95% confidence interval. We also plot the graph search entropy and optimal model size from the real-world FB15K-237 experiment (green dot) to verify the accuracy of the obtained linear scaling law.

number of relations increases the ambiguity in the training set. By adding dummy relations that are not used for reasoning, the language model can better distinguish between the logic rules and spurious correlations between relations. Thus the reasoning performance improves with more relations.

**The optimal model size increases with the deductible ratio when the ratio is small.** In Figure 3 (e), we show the effect of generating knowledge graphs with different ratios between deductible triples and atomic triples, $\gamma$, while keeping the number of entities and the number of triples unchanged. A larger ratio implies that the language model can see more rule patterns at training time, thus improving the reasoning performance. The increase in performance and optimal model size stops after a ratio threshold.

**More entities imply a larger optimal model size.** In Figure 3 (f), we show the effect of generating knowledge graphs with different numbers of nodes/entities $N_e$. In this experiment, we also scale the number of triples to keep all other hyperparameters unchanged. Increasing the number of entities increases the optimal model size while also increasing the testing loss. More entities imply a larger graph which increases the graph complexity, thus increasing the optimal model size. As in this experiment, we use a small number of rule ($N_h = 5$) and relations ($N_r = 10$), an excessive number of entities and triples will create more ambiguity thus hurting the reasoning performance.

## 5.2 OPTIMAL MODEL SIZE V.S. GRAPH SEARCH ENTROPY

From our previous ablation studies, we hypothesize that the optimal model size is positively related to the overall complexity of the knowledge graph. Thus, we propose that we measure the complexity of a knowledge graph by quantifying the amount of information that can be obtained from the graph by exploring the graph through a random search. From our task definition, to reason over the knowledge graph, the language model needs to (a) identify the set of logic rules by observing repetitive patterns; (b) traverse the graph using one or more specific logic rules to locate the tail entity. So we define the **graph search entropy** as the maximum amount of information that can be obtained when randomly traversing the graph.

To simplify the problem, we first focus on the average amount of information we can observe at one node of the graph. If we consider a random walk over the knowledge graph, then we refer to the entropy produced by each step/node on the walk trace for an infinitely long random walk as the *entropy rate* of this random walk. For a graph $G$, the maximum entropy rate is equal to the log of the largest eigenvalue of the adjacency matrix $A$. Note that only consider the entropy rate with respect to the entity, without considering the entropy rate with respect to the relation. We can compute the relation entropy rate with the stationary distribution and transition matrix induced by the maximal entropy rate random walk. If we denote the dominating eigenvalue by $\lambda \in \mathbb{R}$ and the corresponding eigenvector by $\psi \in \mathbb{R}^{N_e}$, then the stationary distribution $\rho \in \mathbb{R}^{N_e}$ can be written as:

$$\rho_i = \psi_i / ||\psi||_2^2.$$

The transition matrix $S \in \mathbb{R}^{N_e \times N_e}$ of the maximal entropy random walk can be written as:

$$S_{ij} = (A_{ij}/\lambda)(\psi_j/\psi_i).$$

We can then transform the entity-to-entity transition matrix $S \in \mathbb{R}^{N_e \times N_e}$ into an entity-to-relation transition matrix $S^r \in \mathbb{R}^{N_e \times N_r}$ by merging the entries with the same relation together:

$$S_{ij}^r = \sum_{k=1}^{N_e} \mathbb{1}[(i, j, k) \in G] S_{ik}.$$

Finally, the relation entropy rate $H^r(G)$ can be written as:

$$H^r(G) = -\sum_{i=1}^{N_e} \rho_i \sum_{j=1}^{N_r} S_{ij}^r \log(S_{ij}^r).$$

The overall **graph search entropy** $H(G)$ can then be written as the sum of the entity entropy rate and the relation entropy rate multiplied by the number of nodes:

$$H(G) = N_e(\log(\lambda) + H^r(G)).$$

We empirically investigate the relation between the optimal model and the graph search entropy by plotting them against each other in Figure 4, and perform linear regression. The optimal model sizes are obtained from the synthetic experiments conducted in the ablation studies. In the ablation studies we only report the results for exponentially increasing model sizes for clarity. In this study to better capture the optimal model size, we make the model sizes near the optimal model size more fine-grain. In all experiments, we keep the training hyperparameter the same, with 10k train steps.

We find a strong linear relation between the optimal model size and the graph search entropy with $R^2 = 0.85$. Note that there are a few sources of noise for locating the optimal model size for a specific knowledge graph. First, we only train language model with selected sizes due to compute and time limitations, and the quantization of the model size would disrupt the smoothness of the scaling law. Second, the exact location of the optimal model size is dependent on the training steps, which we did not thoroughly traverse but choose to inspect at the training step 10k.

After fitting a linear regression line using the data from our synthetic experiments, we check the validity of this empirical scaling law against our real-world knowledge graph, FB15K-237. We calculate the graph search entropy for FB15K-237, and find the predicted optimal model size is very close to the observed optimal model size, shown as a green dot in Figure 4.

From our scaling law, we can see that roughly 124 additional parameters in the optimal model size are required per 1-bit entropy increase in the knowledge graph. That is a language model can only reliably (not perfectly) reason over 0.008 bit information per parameter. This is very different from the knowledge capacity scaling law concluded by Allen-Zhu & Li (2025), which shows that the language model can store 2 bits of knowledge per parameter. We think this discrepancy is due to two reasons: first, our scaling law is not only about memorizing the knowledge, but also about reasoning over the learned knowledge, which is significantly harder. Second, the way we compute the graph search entropy is fundamentally different from the way Allen-Zhu & Li (2025) computes the knowledge entropy. While Allen-Zhu & Li (2025) describes the entropy of the knowledge generation process, our graph search entropy describes the entropy of randomly traversing a fixed knowledge graph. In this way, we did not directly measure the amount of information that a language model needs to memorize, but measuring the complexity of traversing, and therefore, reasoning over a graph. It is hard, if not impossible, to obtain the data generation process of real-world data, but it is possible to get an estimate of the underlying knowledge graph of a corpus through automated knowledge graph construction algorithms (Zhong et al., 2023). Thus, it is possible to predict the optimal reasoning model size for real-world pretraining, by first constructing a knowledge graph from the pretraining corpus, and then computing its graph search entropy, and finally using a similar scaling law to calculate the optimal model size.

## 5.3 LIMITATIONS

We want to highlight that this study is only conducted on simplified pretraining data from knowledge graphs, and the results are not directly applicable to real-world language model pretraining with large

text corpus. The setting of our study provides a reasonable analogy to the real-world language model pretraining, and the obtained insight might be found useful in the real world when the compute is abundant with very large models and very large datasets that exhaustively traverse the underlying knowledge graph. We leave the work of verifying our scaling law in the real word to future research due to its resource-demanding nature.

## 6 RELATED WORK

**Language Model Scaling Laws**  Kaplan et al. (2020) first observed a power-law relationship between LLM perplexity, model parameter count, and training data size, laying the foundation for scaling law research. Subsequently, Hoffmann et al. (2022b) explored optimal training strategies under constrained computational resources and discovered that LLM parameter size and the number of training tokens should scale proportionally to achieve optimal compute efficiency under a fixed budget. Beyond pretraining performance, researchers further confirmed that downstream task performance can also be reliably predicted based on model size and training data volume (Hernandez et al., 2021; Isik et al., 2024). Allen-Zhu & Li (2025); Lu et al. (2024) have turned to exploring more specific capability dimensions, focusing particularly on the scaling laws of factual memory in LLMs and their behavioral patterns when memorizing different types of facts. Most recently, Roberts et al. (2025) have confirmed that scaling laws are skill-dependent, and found that knowledge-intensive tasks are more parameter-hungry while reasoning-intensive tasks are more data-hungry. Springer et al. (2025) challenge a core assumption in scaling research—that more pretraining invariably leads to better downstream performance. Our paper identifies a different U-shaped scaling curve under the specific scenario of knowledge graph reasoning and reveals that the search complexity of the knowledge graph determines the optimal model size. This echoes the discovery of Pandey (2024) and Yin et al. (2024) that classic scaling laws are highly dependent on the data complexity or the compression ratio of the data. Havrilla & Liao (2024) also confirmed from both theoretical and empirical perspectives that the power of the power scaling law depends on the intrinsic dimension of the training data.

**Language Model Reasoning**  Our paper focuses on the reasoning capability of LMs which has drawn a lot of attention recently (Zhang et al., 2023; Chen et al., 2023; Yao et al., 2023a;b; Wang et al., 2023; Guo et al., 2025; Jin et al., 2024; Yeo et al., 2025; Team et al., 2025; Li et al., 2025). LLMs are usually trained to reason in a step-by-step manner in real-world tasks like math problems (Wei et al., 2022b) and coding (Yang et al., 2024). In our experiments, we do not ask LMs to generate a CoT solution, but ask the language model to directly choose the correct answer from the given options, because our pretrain-only LMs are not trained to give a CoT solution for a query. Our synthetic reasoning environment is the most similar to Wang et al. (2024b), which also use the knowledge graph completion task as a testbed to understand how LMs learn to reason at pretraining time. They propose that LMs are able to aggregate random walk paths sampled from the knowledge graph. Wang et al. (2024a); Zhu et al. (2024) also employ a graph structure to ground their synthetic reasoning tasks to explain how LLMs reason, but their reasoning is defined as concatenations of relations: A is $r_1$ to B and B is $r_2$ to C implies A is $r_1 r_2$ to C. The knowledge graph completion task we employ is more complex than simple concatenation of relations as the language model needs to find out which relation $r_1 r_2$ corresponds to from the knowledge graph.

**Science of Language Model Reasoning**  Several recent works have advanced the understanding of reasoning in large language models by investigating how scaling, fine-tuning, and reinforcement learning affect their capabilities. Zhang et al. (2024) systematically examine the interplay of model size, pretraining data, fine-tuning data, and tuning methods, finding that fine-tuning performance follows a power-law scaling with data and model size. Zhao et al. (2025) focus on reinforcement learning (RL) based post-training and observe that RL drives models toward a single dominant output distribution, effectively amplifying patterns already present in the pretraining data. Qi et al. (2025) introduce EvoLM, a comprehensive framework to analyze training dynamics across all stages (from pre-training to RL fine-tuning), and report diminishing returns from excessively long pre-training or post-training while highlighting the crucial role of an intermediate continued-pretraining phase to prevent knowledge forgetting and better bridge the gap between base model pretraining and downstream fine-tuning. Finally, Yue et al. (2025) show that current RL-with-verifiable-reward methods do not elicit fundamentally new reasoning abilities beyond what the base model already possesses. In their experiments, RL-finetuned models outperformed the base model on strict one-

answer evaluations (e.g. pass@1), but the base model achieved higher success when more attempts were allowed (large pass@k), implying that RL primarily exploits existing reasoning patterns rather than creating novel ones. This line of work explore the science of language model reasoning from a more empirical perspective which is complementary to our findings.

## 7 CONCLUSION

This paper presents a rigorous study of the scaling behavior of implicit reasoning in language models pretrained on knowledge graphs. Our findings reveal a U-shaped relationship between implicit reasoning performance and model size: overparameterization degrades reasoning ability. We further identify key factors that determine the optimal model size, including the number of training triples and the complexity of the graph. Most notably, we propose an empirical scaling law that links the optimal model size to graph search entropy, demonstrating that a language model can reason over approximately 0.008 bits of information per parameter. Although our experiments are conducted in controlled settings to ensure rigor, the insights derived from this work offer promising directions for future studies on real-world pretraining and the enhancement of reasoning capabilities in large language models.

## 8 REPRODUCIBILITY STATEMENT

We have taken several steps to ensure the reproducibility of our results. All experimental settings, including model architectures, training procedures, and hyperparameters, are described in detail in Section 2, Section 5 and Appendix B. To facilitate empirical reproducibility, we include a script of data construction steps in Appendix D. Additionally, we provide a simplified version of our source code as an easy-to-run Jupyter notebook for reproducing some of our experiments in the supplementary materials.

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

APPENDIX

# A  DATA PROCESSING EXAMPLE

| Type | Example |
|------|---------|
| Original | "triple": {
    "head": "drama film",
    "relation": "/media_common/netflix_genre/titles",
    "tail": "American History X"
} |
| GPT4 generated | "The drama film includes \"American History X\" as one of its Netflix genre titles." |
| Template | "template": "$tail was released as part of the $head genre on Netflix during its period of popularity.",
"sentence": "American History X is featured under the drama film genre on Netflix." |
| Triple-only | (1254, 22, 765) |

Figure 5: An example of a triple being processed in three different ways.

# B  EXPERIMENT DETAILS

| Model size | hidden size | MLP size | #attention heads | #layers |
|:----------:|:-----------:|:--------:|:----------------:|:-------:|
| 0.3M | 128 | 256 | 2 | 2 |
| 0.7M | 128 | 256 | 2 | 4 |
| 1.3M | 256 | 512 | 4 | 2 |
| 2.6M | 256 | 512 | 4 | 4 |
| 5.3M | 256 | 512 | 4 | 8 |
| 10.5M | 512 | 1024 | 8 | 4 |
| 21.0M | 512 | 1024 | 8 | 8 |
| 42.0M | 512 | 1024 | 8 | 16 |
| 83.9M | 1024 | 2048 | 16 | 8 |
| 167.8M | 1024 | 2048 | 16 | 16 |
| 335.6M | 1024 | 2048 | 16 | 32 |
| 671.2M | 2048 | 4096 | 32 | 16 |
| 1342.4M | 2048 | 4096 | 32 | 32 |

Table 1: Language model (Llama) size details

| batch size | lr | lr scheduler | warmup ratio | weight decay | max length |
|:----------:|:---:|:-----------:|:------------:|:------------:|:----------:|
| 1024 | 1e-4 | cosine | 0.2 | 0 | 128 |

Table 2: Hyperparameter settings for language model pretraining.

|     | $N$ | $N_e$ | $N_r$ | $N_h$ | $\gamma$ |
|-----|-----|-------|-------|-------|----------|
| (a) | 100k | 10k | 100 | 50 | 0.5 |
| (b) | 10k/20k/.../100k | 10k | 100 | 50 | 0.5 |
| (c) | 100k | 10k | 100 | 5/10/.../50 | 0.5 |
| (d) | 100k | 10k | 10/20/.../100 | 50 | 0.5 |
| (e) | 100k | 10k | 100 | 50 | 0.1/0.5/.../0.9 |
| (f) | 10k/20k/.../100k | 1k/2k/.../10k | 10 | 5 | 0.5 |

Table 3: Knowledge graph hyperparameter settings for Figure 3 experiments. We keep $L_{min} = 2$ and $L_{max} = 4$ for all experiments. Here $N$ denotes the number of triples, $N_e$ denotes the number of entities, $N_r$ denotes the number of relations, $N_h$ denotes the number of rules, $\gamma$ denotes the ratio between deductible triples and atomic triples, $L_{min}$ denotes the minimum rule length, and $L_{max}$ denotes the maximum rule length.

## C  SYNTHETIC KNOWLEDGE GRAPH V.S. REAL-WORLD KNOWLEDGE GRAPH

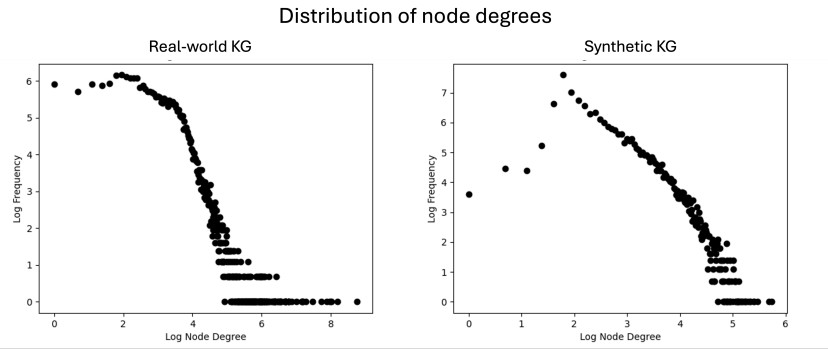

Figure 6: Distribution of node degrees of synthetic and real-world knowledge graphs.

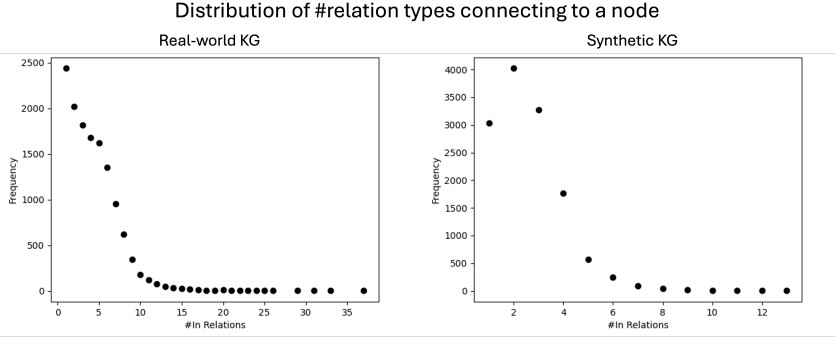

Figure 7: Distribution of number of outgoing relations per node of synthetic and real-world knowledge graphs.

## D  SYNTHETIC KNOWLEDGE GRAPH GENERATION CODE

```python
import networkx as nx
import numpy as np
import random
from collections import defaultdict

def add_edge(G, h, t, r):
    num_edges = 0
    if G.has_edge(h, t):
        if r not in G[h][t]['id']:
            G[h][t]['id'].append(r)
            num_edges += 1
        else:
            print('edge already exists')
    else:
        G.add_edge(h, t, id=[r])
        num_edges += 1
    print('add edge: ', (h, r, t), 'num edges: ', num_edges)
    return num_edges

def generate_rules(relations, num_rules, L_min, L_max, weighted=False, temperature=0.25):
    # Generate K acyclic logic rules with varying lengths
    dependency_graph = defaultdict(set)
    rules = []
    weights = []
    if weighted:
        for l in range(L_min, L_max + 1):
            weights.append(np.exp(-temperature*l))
        probs = np.array([w / sum(weights) for w in weights])
    else:
        weights = [1] * (L_max - L_min + 1)

    def has_cycle(start, visited, stack):
        """Detects if adding a new dependency introduces a cycle."""
        if start not in visited:
            visited.add(start)
            stack.add(start)
            print('visited: ', visited)
            print('stack: ', stack)
            for neighbor in dependency_graph[start]:
                if neighbor in stack:
                    return True
                elif has_cycle(neighbor, visited, stack):
                    return True
        if start in stack:
            stack.remove(start)
        return False

    for _ in range(num_rules):
        while True:
            if weighted:
                length = random.choices(range(L_min, L_max + 1), weights=weights)[0]
            else:
                length = random.randint(L_min, L_max)
            rule_relations = random.choices(relations, k = length + 1) # the first element is the implied relation
            valid_rule = True
            for i in range(1, len(rule_relations)):
```

```
864                         dependency_graph[rule_relations[0]].add(rule_relations[i])
865

866                         # Check for cycles
867                         if has_cycle(rule_relations[i], set(), set()):
868                             valid_rule = False
869                             for j in range(1, i + 1):
870                                 dependency_graph[rule_relations[0]].remove(rule_relations[j])
871                             break
872
873                 if valid_rule:
874                     rules.append(tuple(rule_relations))
875                     break
876
877     print('rules: ', rules)
878     return rules
879
880 def get_node_types(rules, max_num_relations_per_node=3):
881     # map node types to out relations
882     node_types = {}
883     # map out relations to node types
884     r2node_types = defaultdict(list)
885     for rule in rules:
886         for i in range(len(rule)):
887             node_type = len(node_types)
888             if i == 0:
889                 node_types[node_type] = [rule[i], rule[1]]
890                 r2node_types[rule[i]].append(node_type)
891                 r2node_types[rule[1]].append(node_type)
892             elif i == len(rule) - 1:
893                 node_types[node_type] = ['-' + rule[i], '-' + rule[0]]
894                 r2node_types['-' + rule[i]].append(node_type)
895                 r2node_types['-' + rule[0]].append(node_type)
896             else:
897                 node_types[node_type] = ['-' + rule[i], rule[i+1]]
898                 r2node_types['-' + rule[i]].append(node_type)
899                 r2node_types[rule[i+1]].append(node_type)
900
901     print(node_types)
902     print(r2node_types)
903
904     for num_rs in range(2, max_num_relations_per_node):
905         possible_new_node_types = []
906         for r in r2node_types:
907             alt_rs = []
908             for node_type in r2node_types[r]:
909                 for _r in node_types[node_type]:
910                     if _r != r:
911                         alt_rs.append(_r)
912             alt_rs = list(set(alt_rs))
913             for node_type in r2node_types[r]:
914                 if len(node_types[node_type]) == num_rs:
915                     for _r in alt_rs:
916                         if _r not in node_types[node_type]:
917                             possible_new_node_types.append(tuple(sorted([_r] + list(node_types[node_type]))))
            print(possible_new_node_types)
            possible_new_node_types += list(set(possible_new_node_types))
        possible_new_node_types = list(set(possible_new_node_types))
        print(possible_new_node_types)
```

```
918             for rs in possible_new_node_types:
919                 new_node_type = len(node_types)
920                 node_types[new_node_type] = list(rs)
921                 for _r in rs:
922                     r2node_types[_r].append(new_node_type)
923
924         return node_types
925
926     def get_adj_out_relations(rules):
927         adj = defaultdict(list)
928         for rule in rules:
929             for i in range(len(rule)):
930                 if i == 0:
931                     adj[rule[i]].append(rule[1])
932                     adj[rule[1]].append(rule[i])
933                 elif i == len(rule) - 1:
934                     adj['-' + rule[i]].append('-' + rule[0])
935                     adj['-' + rule[0]].append('-' + rule[i])
936                 else:
937                     adj['-' + rule[i]].append(rule[i+1])
938                     adj[rule[i+1]].append('-' + rule[i])
939         return adj
940
941     def latent_rule_graph(num_rules=50, L_min=2, L_max=4, n=10000, m=10, n_r=200,
942                           num_test=1000, num_train=150000, check_frequency=100,
943                           power_law=False, initial_graph=None,
944                           length_weighted=False, mcmc=0.2, temperature=0.25,
945                           deductible_ratio=0.5):
946         # Generate relations and entities
947         print("mcmc: ", mcmc)
948         relations = ['P' + str(i) for i in range(n_r)]
949         all_rules = generate_rules(relations, max(n_r//L_min, num_rules), L_min, L_max)
950         r2rules = {}
951         for rule in all_rules:
952             if rule[0] not in r2rules:
953                 r2rules[rule[0]] = []
954             r2rules[rule[0]].append(rule[1:])
955         num_triples = 0
956         repeated_entities = defaultdict(list) # map in relation to entities
957         child_relations = []
958         for rule in all_rules:
959             child_relations += rule[1:]
960         child_relations = list(set(child_relations))
961         child_relations += ['-' + r for r in child_relations]
962         deductible_rules = random.sample(all_rules, num_rules)
963         if length_weighted:
964             weights = [int(100*np.exp(-temperature*len(rule))) for rule in all_rules]
965         else:
966             weights = [1 for _ in all_rules]
967         repeated_rules = []
968         for rule, weight in zip(all_rules, weights):
969             for _ in range(weight):
970                 repeated_rules.append(rule)
971         random.shuffle(repeated_rules)
        adj = get_adj_out_relations(repeated_rules)
        all_deductibles = {}

        if initial_graph is None:
            # Default initial graph
```

```
972             G = nx.DiGraph()
973             node_id = 0
974             min_repeated_entities = 0
975             while min_repeated_entities < m:
976                 for rule in all_rules:
977                     source = 'Q' + str(node_id)
978                     node_id += 1
979                     h = source
980                     for r in rule[1:]:
981                         t = 'Q' + str(node_id)
982                         node_id += 1
983                         num_triples += add_edge(G, h, t, r)
984                         repeated_entities[r].append(t)
985                         repeated_entities['-' + r].append(h)
986                         h = t
987                     num_triples += add_edge(G, source, t, rule[0])
988                     repeated_entities[rule[0]].append(t)
989                     repeated_entities['-' + rule[0]].append(source)

990                 min_repeated_entities = min([len(set(repeated_entities[r])) for r in child_relations])
991         else:
992             if len(initial_graph) < m or len(initial_graph) > n:
993                 raise nx.NetworkXError(
994                     f"Initial graph needs between m={m} and n={n} nodes"
995                 )
996             G = initial_graph.copy()
997             node_id = len(G)

998         if not power_law:
999             repeated_entities = {r: list(set(repeated_entities[r])) for r in repeated_entities}

1000
1001        # Start adding the other nodes.
1002        while node_id < n:
1003            source = 'Q' + str(node_id)
1004            node_id += 1
1005            possible_relations = [_r for _r in adj if _r in child_relations]
1006            if len(possible_relations) == 0:
1007                print('no adj relations')
1008                break
1009            print('add child edge')
1010            chosen_edges = []
1011            stop = False
1012            for _ in range(m):
1013                it = 0
1014                while (r, t) in chosen_edges:
1015                    r = random.choice(possible_relations)
1016                    t = random.choice(repeated_entities[r])
1017                    it += 1
1018                    if it > 100:
1019                        print('failed to find edge')
1020                        stop = True
1021                        break
1022                if stop or len(possible_relations) == 0:
1023                    break

1024                possible_relations = [_r for _r in adj[r] if _r in child_relations]
1025                chosen_edges.append((r, t))
1026                if r[0] == '-':
1027                    num_triples += add_edge(G, t, source, r[1:])
```

```
1026                        repeated_entities[r[1:]].append(source)
1027                else:
1028                    num_triples += add_edge(G, source, t, r)
1029                    repeated_entities['-' + r].append(source)
1030                repeated_entities[r].append(t)
1031                if len(possible_relations) == 0:
1032                    print('no adj relations')
1033                    break
1034
1035        if not power_law:
1036            repeated_entities = {r: list(set(repeated_entities[r])) for r in repeated_entities}
1037
1038        if node_id % check_frequency == 0 or node_id == n-1:
1039            # add deductibles
1040            all_nodes = list(G.nodes)
1041            random.shuffle(all_nodes)
1042            for h in all_nodes:
1043                for rule in deductible_rules:
1044                    head_list = [h]
1045                    r = rule[0]
1046
1047                    for _r in rule[1:]:
1048                        next_head_list = []
1049                        for e_h in head_list:
1050                            if e_h not in G.nodes:
1051                                continue
1052                            for e_t in G[e_h]:
1053                                if _r in G[e_h][e_t]['id']:
1054                                    if random.random() < mcmc:
1055                                        next_head_list.append(e_t)
1056                        head_list = next_head_list
1057
1058                    for t in head_list:
1059                        if (h, r, t) not in all_deductibles:
1060                            all_deductibles[(h, r, t)] = [rule]
1061                        elif rule not in all_deductibles[(h, r, t)]:
1062                            all_deductibles[(h, r, t)].append(rule)
1063                        if not G.has_edge(h, t) or r not in G[h][t]['id']:
1064                            print('add deductible edge')
1065                            add_edge(G, h, t, r)
1066                            num_triples += 1
1067                            repeated_entities[r].append(t)
1068                            repeated_entities['-' + r].append(h)
1069
1070    atomic_triples = []
1071    deductible_triples = []
1072    for h, t in G.edges:
1073        for r in G[h][t]['id']:
1074            if (h, r, t) not in all_deductibles:
1075                atomic_triples.append((h, r, t))
1076            else:
1077                deductible_triples.append((h, r, t))
1078    random.shuffle(atomic_triples)
1079    random.shuffle(deductible_triples)
     assert len(atomic_triples) >= int(num_train * (1-deductible_ratio))
     assert len(deductible_triples) >= int(num_train * deductible_ratio) + 2 * num_test

     remove_triples = []
     train_atomic_triples = atomic_triples[:int(num_train * (1-deductible_ratio))]
```

```
1080    remove_triples += atomic_triples[int(num_train * (1-deductible_ratio)):]
1081    train_deductible_triples = deductible_triples[:int(num_train * deductible_ratio)]
1082    remove_triples += deductible_triples[int(num_train * deductible_ratio):]
1083
1084    for h, r, t in remove_triples:
1085        _t = t
1086        rs = G[h][_t]['id']
1087        if r in rs:
1088            if len(rs) == 1:
1089                G.remove_edge(h, _t)
1090            else:
1091                G[h][_t]['id'].remove(r)
1092
1093    train_triples = train_deductible_triples + train_atomic_triples
1094    random.shuffle(train_triples)
1095    print("num train triples: ", len(train_triples))
1096
1097    r2rule = {}
1098    for rule in deductible_rules:
1099        if rule[0] in r2rule:
1100            r2rule[rule[0]].append(rule[1:])
1101        else:
1102            r2rule[rule[0]] = [rule[1:]]
1103
1104    def check_deductible(triple):
1105        h, r, t = triple
1106        alt_ts = []
1107        for rule in r2rule[r]:
1108            head_list = [h]
1109            for _r in rule:
1110                next_head_list = []
1111                for e_h in head_list:
1112                    for e_t in G[e_h]:
1113                        if _r in G[e_h][e_t]['id']:
1114                            next_head_list.append(e_t)
1115                head_list = next_head_list
1116            alt_ts += head_list
1117        if t in alt_ts:
1118            return True
1119        return False
1120
1121    id_test_triples = []
1122    for i in range(int(num_train * deductible_ratio), len(deductible_triples)):
1123        if check_deductible(deductible_triples[i]):
1124            id_test_triples.append(deductible_triples[i])
1125        if len(id_test_triples) == num_test:
1126            break
1127
1128    id_test_rules = [all_deductibles[triple] for triple in id_test_triples]
1129    print("num id test triples: ", len(id_test_triples))
1130
1131    rule2triples = defaultdict(list)
1132    for triple in deductible_triples[i+1:]:
1133        for rule in all_deductibles[triple]:
1134            rule2triples[rule].append(triple)

    # uniformly sample testing triples from each rule
    uniform_test_triples = []
    for rule in rule2triples:
```

```
        triples = []
        for triple in rule2triples[rule]:
            if check_deductible(triple):
                triples.append(triple)

        if len(triples) > num_test//len(rule2triples):
            uniform_test_triples += random.sample(triples, num_test//len(rule2triples))
        else:
            uniform_test_triples += triples

    random.shuffle(uniform_test_triples)
    uniform_test_rules = [all_deductibles[triple] for triple in uniform_test_triples]
    print("num uniform test triples: ", len(uniform_test_triples))

    return G, deductible_rules, train_triples, id_test_triples, id_test_rules, uniform_test_triples, uniform_test_rules
```

