# OpenReview forum: "Do Larger Language Models Generalize Better? A Scaling Law for Implicit Reasoning at Pretraining Time"
_ICLR.cc/2026/Conference — Submitted to ICLR 2026_

### Official Review · Reviewer_7rMj · 2025-11-02

**Soundness:** 2
**Presentation:** 2
**Contribution:** 2
**Rating:** 4
**Confidence:** 3

**Summary:**

The authors seek to derive a scaling law for implicit reasoning performance compared to model size by pretraining models on real and synthetic knowledge graphs. They observe a U-shaped relationship between model-size and implicit reasoning performance (measured on a held out set of knowledge graphs) and conclude that larger models “memorize” instead of reasoning on the knowledge gained from graphs, hurting test-time performance. The authors propose an empirical scaling law that demonstrates the optimal model size compared to “graph search entropy,” a metric for the complexity of a given graph.

**Strengths:**

-	The authors introduce a novel, controlled testbed for measuring reasoning performance and connect the synthetic knowledge graph generation to realistic knowledge graph datasets.
-	The authors are very thorough in ablating over relevant parameters in section 5.1
-	The paper introduces graph search entropy as a way to capture the complexity of a graph and demonstrates how the optimal model size is correlated with this metric
-	They release code for reproducibility

**Weaknesses:**

-	I have some concerns over how generalizable these findings are. First, the experiments are all run on knowledge graphs with artificial logic rules. The authors acknowledge that these findings may not carry over to natural language reasoning. Furthermore, knowledge graph completion seems very different from reasoning in the real world, particularly when the benchmark is multiple choice. The 0.008 bits/parameter claim seems misleading. Finally, the authors seem to suggest that memorization hurts reasoning performance but don’t clearly show how this is the case. Some analysis on how the large and small models are learning/reasoning differently would be interesting
-	The experiment settings are odd in that all models, regardless of size, are trained for the same number of examples and steps rather than controlling for flops or tokens per parameters. Additionally, the U-shaped test loss compared to model size suggests that the overparametrized models are overfitting to the training data. Is this the case, and how realistic is scenario in large language models? The FB15K-237 experiments are run with 30 data repeats.
-	The graph search entropy is a novel and creative way to illustrate the graph complexity. However, I would appreciate if the authors can provide some more intuition on how this is derived.

**Questions:**

Do the U shaped curves persist when controlling for flops/data, particularly when reducing the number of data repeats?

Is there a way to demonstrate that models are learning logical rules about the knowledge graphs and that the “optimal size” models are learning differently?

Does the optimal size change when altering the dataset (ie increasing/decreasing the number of answer choices

---

> ### Author Response · Authors · 2025-11-26
>
> We thank the reviewer for their feedback. Below is our response:
>
> 1. Concerns on experiment setting: see general response point 1.
>
> 2. “authors seem to suggest that memorization hurts reasoning performance but don’t clearly show how this is the case”:  This only appears ONCE in the abstract as an intuitive introduction to our observation and never appears as a formal claim in the main text. Evidence for this intuitive sentence: the training loss is able to monotonically decrease to a very small value when model size becomes larger as shown in figure 1 (c)(f)(i), while the testing loss shows an U-shape. Such an effect is also observed with synthetic KG, but due to the highly repetitive pattern, we didn’t include them in the paper. We have removed the word “memorization” from the abstract.
>
> 3. “The experiment settings are odd in that all models, regardless of size, are trained for the same number of examples and steps rather than controlling for flops or tokens per parameters.”: Please refer to line 193 and 264-269. We observe that the optimal model size and the maximum accuracy/minimum testing loss  is stable when the training step is large enough. So we choose to repeat the training data many times. We have made it clearer in the paper.
>
> 4. “The graph search entropy is a novel and creative way to illustrate the graph complexity. However, I would appreciate if the authors can provide some more intuition on how this is derived.”: Please refer to line 363-377. “we define the graph search entropy as the maximum amount of information that can be obtained when randomly traversing the graph.”
>
> 5. “Do the U shaped curves persist when controlling for flops/data, particularly when reducing the number of data repeats?”: U shape is most significant when training for larger number of steps.. See general response point 3 for more discussion.
>
> 6. “Is there a way to demonstrate that models are learning logical rules about the knowledge graphs and that the “optimal size” models are learning differently?”: The model with the optimal model size does not learn differently. It just learns better. For the exact learning mechanism, please refer to [1]. They use almost identical experimental settings of implicit reasoning and show that LMs aggregate paths from KG seen at pretraining time to infer unseen triples.
>
> 7. “Does the optimal size change when altering the dataset (ie increasing/decreasing the number of answer choices”: No. Because the optimal model size is with respect to the lowest testing loss, which is independent of the number of answer choices. The multiple choice setup is exclusively used for computing accuracy.
>
> [1] Understanding Reasoning Ability of Language Models From the Perspective of Reasoning Paths Aggregation. Xinyi Wang, Alfonso Amayuelas, Kexun Zhang, Liangming Pan, Wenhu Chen, William Yang Wang. ICML 2024.

---

### Official Review · Reviewer_1bdq · 2025-11-03

**Soundness:** 1
**Presentation:** 2
**Contribution:** 2
**Rating:** 4
**Confidence:** 3

**Summary:**

This paper investigates whether scaling up language models inherently improves their reasoning ability during pre-training. The authors design a controlled setting in which models are trained on structured knowledge graph data represented as triples ($e_1$, r, $e_2$), where entity $e_1$ has relation r with entity $e_2$. Through systematic experiments, they find that reasoning performance does not follow the typical monotonic power-law scaling trend observed for loss or accuracy. Instead, the results exhibit a U-shaped curve, suggesting that beyond a certain size, larger models may over-memorize the training data, which ultimately hinders their reasoning capability.

Overall, the paper tackles an important question about how reasoning ability scales when language models are pretrained on structured data, examining the effects of model size, dataset size, and training steps. However, the study would be stronger with experiments involving larger models and more extensive training data. It is also unclear whether the conclusions drawn from relatively small models can generalize to larger-scale settings. Moreover, according to the appendix, only a single hyperparameter configuration is used across different model sizes, which raises concerns about the robustness of the study.

**Strengths:**

1. The paper addresses a fundamental and timely question, whether scaling up language models inherently enhances reasoning ability during pre-training. The topic is both conceptually important and empirically under explored.
2. The paper presents its methodology and results clearly, with systematic comparisons across model sizes, dataset scales, and training steps, making it relatively easy to reproduce and extend.

**Weaknesses:**

1. the paper could be stronger with experiments involving larger models and more extensive training data.
2. It is also unclear whether the conclusions drawn from relatively small models (like 0.3M model) is safe.
3. Moreover, according to the appendix, only a single hyperparameter configuration is used across different model sizes.

**Questions:**

1. What is the motivation to eliminate the multiple-answer questions? Is it convincing to draw the conclusion only from single answer question?
2. Can authors provide more details about the test set? Is the test set simply constructed by the same distribution of the training knowledge graph? Does the test set cover complex reasoning relationship?
3. I am curious why we need (why we are exploring) a specialized large language model for knowledge graph data. Is the regular LLM reasoning not as good as LLM trained on KG data?

---

> ### Author Response · Authors · 2025-11-26
>
> We thank the reviewer for their feedback. Below is our response:
>
> 1. “only a single hyperparameter configuration is used across different model sizes”: no significant difference has been observed in terms of the optimal model size across different hyperparameters.
>
> 2. “What is the motivation to eliminate the multiple-answer questions?”: Multiple-answer questions will have different loss scales compared to single-answer questions (i.e. the log likelihood would distribute over several answers instead of concentrating on one), thus will disturb the scaling law.
>
> 3. “Can authors provide more details about the test set?”: In line 94-97, we explain that "We test the reasoning capability of a language model trained on such a corpus by testing its accuracy in completing triples that have never been seen in the knowledge graph but can be deduced through latent rules encoded in the graph structure". So no the testing data is not in the same distribution as training data. For synthetic KG, "We also further ensure that the triples in the held-out test set are all deductible through the training triple". So yes, the test set covers all latent logic rules that we used to build the graph, which involve 2-4 hops multihop reasoning.
>
> 4. “I am curious why we need (why we are exploring) a specialized large language model for knowledge graph data.”: The experiment set up is not chosen because of performance, but because of analytical rigor. LM pretrained on large corpus would perform much better on benchmarks, but it doesn't cleanly separate reasoning with knowledge, math computation, linguistic rules, language generation, etc. Different types of reasoning can also exhibit different properties and behavior. We construct a synthetic testbed to exclusively study the case of implicit reasoning. Also see general response point 1.

---

### Official Review · Reviewer_xsY7 · 2025-11-07

**Soundness:** 3
**Presentation:** 2
**Contribution:** 2
**Rating:** 6
**Confidence:** 2

**Summary:**

The paper studies how language model size affects implicit multi-hop reasoning during pretraining using knowledge graph based data, specifically FB15K-237. It defines implicit reasoning as inferring missing edges via multi-hop rules without chain-of-thought supervision and observes a U-shaped relationship between model size and reasoning performance. Overly large models tend to overfit and memorize, degrading implicit reasoning, while the minimum achievable reasoning loss is largely determined by the training data. The authors further show that the optimal model size depends on knowledge graph complexity and introduce a graph search entropy metric to predict it.

**Strengths:**

The paper addresses scaling and reasoning in LLM pretraining, an important yet underexplored problem. It introduces a set of novel experiments that make implicit multi-hop reasoning measurable. The large-scale experiments and the proposed link between knowledge graph complexity, optimal model size, and graph search entropy provide an interpretable and potentially useful perspective.

**Weaknesses:**

1. The training objective is aside from test accuracy and test loss, which might cause unfairness of the scaling law, which may weaken the validity of the proposed scaling law. Also, not sure if the author did the instruction tuning for the multi-choice question answering.
2. Figure 2 requires more explanation; for example, it is unclear why the relation from type2 to type3 uses the same relation r2 as from type5 to type8.
3. The empirical study relies solely on FB15K-237 despite the availability of many standard KG benchmarks. For the sake of generality, at least one additional dataset should be included.

**Questions:**

1. The paper claims a scaling law for LLM pretraining. How did authors formulate the actual training loss in the common causal next-token prediction setting? Especially for the GPT-style textualization where entities and relations are converted to natural language. How is the training objective implemented?
2. Are any general text corpora or external datasets mixed into pretraining, or is training performed exclusively on the constructed knowledge graph data?
3. Is test accuracy evaluated directly after pretraining, or after any additional instruction tuning or adaptation stages?
4. See (1254, 22, 765) in the triple-only example: is it entity and relation id would be tokenized like a natural text?

---

> ### Author Response · Authors · 2025-11-26
>
> We thank the reviewer for their feedback. Below is our response:
>
> 1. “The training objective is aside from test accuracy and test loss” “How did authors formulate the actual training loss in the common causal next-token prediction setting?”: The training loss is the same as the testing loss: next token prediction. We have made it clearer in the paper.
>
> 2. “not sure if the author did the instruction tuning for the multi-choice question answering.”: No instruction tuning is performed. The testing data is in the exact same format as training data.
>
> 3. “for example, it is unclear why the relation from type2 to type3 uses the same relation r2 as from type5 to type8.”: Different logic rules can involve the same relation, as long as they don't form any cycle. This is common in real word KGs: A is B's parent, B is C's parent -> A is C's grandparent.  A is C's grandparent, D is C's daughter, D is not A's parent -> D is C's aunt. Here the relation "grandparent" is reused in the same way as figure 2.
>
> 4. “The empirical study relies solely on FB15K-237 despite the availability of many standard KG benchmarks”: FB15K-237 is the most natural one as it comes from Wikidata and it fits our goal the best. Most other KG benchmarks are domain specific: UMLS is focused on medical domain,  Countries only contains “is neighbor of” and “located in” relationships.  We are trying to mimic real-world knowledge distribution, instead of knowledge in a specific domain. We have considered using WikiData5M but it exceeds our compute budget as it is very large.
>
> 5. “Are any general text corpora or external datasets mixed into pretraining”: No.
>
> 6. “See (1254, 22, 765) in the triple-only example: is it entity and relation id would be tokenized like a natural text?”: They are tokenized per character with our own tokenizer as stated in line 173.

---

### Official Review · Reviewer_nSeB · 2025-11-08

**Soundness:** 2
**Presentation:** 2
**Contribution:** 2
**Rating:** 2
**Confidence:** 4

**Summary:**

This paper presents an empirical finding that language models trained on synthetic multi-hop knowledge graph data demonstrate non-monotonic performance with increasing model size. Across both semi-synthetic (using a real knowledge graph) and fully synthetic (random graph) experiments, a similar trend is observed and a scaling law is fit to the optimal model size as a function of the "graph search entropy".

**Strengths:**

1. The paper presents an interesting and counterintuitive result of a particular task that may not scale with model capacity.

2. There are careful synthetic experiments which present a simple model that can capture a lot of the effect observed in the knowledge graph case.

**Weaknesses:**

1. Some of the claims about models learning reasoning vs. memorization do not seem well substantiated. For example it is brought up in the abstract that large models are failing due to excessive memorization, but there is no clear evidence of this in any of the experiments (which only track loss/accuracy, and don't really try to define or measure memorization). There is also maybe some missing related work on the subject, e.g. this paper that studies memorization vs. reasoning in MoEs: https://arxiv.org/abs/2410.19034.

2. It is not clear how much the severe epoching is changing the results. I understand that as the paper says, facts may be presented multiple times, but in those cases it will always be in different contexts and with different phrasing (especially in deduplicated data). You can see this effect in figure 1. For the GPT-generated and templated data (which contain some variation in phrasing), when the number of steps is low (i.e. less epoching), then scaling the model is generally better. It is only in the cases of extreme over-training and insufficient rephrasing where the overfitting effect is observed. In this way, it is not clear if this is picking up a real effect or just showing that with enough epoching, overfitting is indeed possible.

3. In general, it is not clear how the semi-synthetic experiments connect to real world language modeling.

**Questions:**

1. Is there an experiment you could do to substantiate the memorization claims?

2. Is there an experiment you could do to better test the epoching claims? Can you also change the plots to be presented in terms of epochs instead of "steps" which is sort of a meaningless metric since it also depends on batch size?

3. Is there an eval that could be run on real language model training that could observe the same effect?

---

> ### Author Response · Authors · 2025-11-26
>
> We thank the reviewer for their feedback. Below is our response:
>
> 1. “Some of the claims about models learning reasoning vs. memorization do not seem well substantiated”: This only appears ONCE in the abstract as an intuitive introduction to our observation and never appears as a formal claim in the main text. Evidence for this intuitive sentence: the training loss is able to monotonically decrease to a very small value when model size becomes larger as shown in figure 1 (c)(f)(i), while the testing loss shows an U-shape. Such an effect is also observed with synthetic KG, but due to the highly repetitive pattern, we didn’t include them in the paper. We have removed the word “memorization” from the abstract.
>
> 2. “It is not clear how much the severe epoching is changing the results.”: with a smaller number of training steps, it is obvious that large models would perform better than smaller models, and this has been shown in numerous studies. What we want to argue is that under appropriate training settings (more training epochs), small models can perform as well as large models (see figure 1(g)). Why do we want to study the setting where a smaller model performs well while a large model performs bad? First, we want to test the limit of reasoning capacity of language models: what is the smallest number of parameters that can reach the maximum reasoning performance? Second, larger models need more resources to train and serve. From an efficiency perspective, we want to always use the smallest model possible to reach the same performance.
>
> 3. “Can you also change the plots to be presented in terms of epochs instead of "steps" which is sort of a meaningless metric since it also depends on batch size?”: The batch size is constant, so the step number is directly comparable. The number of epochs we use is not an integer so we show step number instead.
>
> 4. “Is there an eval that could be run on real language model training that could observe the same effect?”: see general response point 1 and 2.

---

### Official Review · Reviewer_CJHz · 2025-11-12

**Soundness:** 3
**Presentation:** 2
**Contribution:** 3
**Rating:** 6
**Confidence:** 4

**Summary:**

This paper explores how the scaling of language model size affects reasoning capabilities that emerge implicitly during pretraining, without explicit chain-of-thought supervision. The authors pretrain language models from scratch on both real-world and synthetic knowledge graphs, where reasoning is operationalized as predicting missing edges that require multi-hop inference. Contrary to the conventional belief that larger models always perform better, the study finds a U-shaped relationship between model size and reasoning ability: models that are too large exhibit degraded reasoning performance due to overparameterization and memorization.

Through systematic experiments on synthetic graphs, the paper shows that the optimal model size depends on properties of the data, such as the number of entities, relations, and triples, but not on the number of training steps. The authors propose a new quantitative measure called graph search entropy, which captures the information-theoretic complexity of reasoning paths in a graph. They then establish an empirical scaling law linking the optimal model size to this entropy measure and show that, on average, a model can effectively reason over about 0.008 bits of information per parameter. This finding contrasts sharply with prior work on knowledge memorization capacity, which estimated around 2 bits per parameter.

Overall, the work presents a controlled experimental framework for studying implicit reasoning during pretraining, reveals a non-monotonic scaling law that challenges standard assumptions about model size and performance, and introduces a theoretical construct that provides a new lens for understanding the relationship between reasoning, memorization, and scale in language models.

**Strengths:**

This paper is original in its problem formulation and experimental design. Rather than following the conventional view that larger language models always perform better, it investigates how reasoning abilities emerge during pretraining and demonstrates a striking U-shaped relationship between model size and reasoning performance. The proposed notion of implicit reasoning framed through knowledge graph completion provides a controlled and interpretable setting that isolates reasoning from linguistic or task-specific confounds. This conceptual reframing, along with the introduction of the graph search entropy measure, represents a creative and meaningful advance in understanding the link between data complexity and model capacity.

The work is of good quality, supported by experiments on synthetic datasets. The methodology is thoughtfully constructed, the results are reproducible, and the analyses are clear and consistent. The discovery that optimal reasoning capacity scales linearly with graph entropy and amounts to about 0.008 bits per parameter adds a precise, quantitative dimension to the study, distinguishing it from prior scaling law research.

The paper is clearly written and well-organized, with figures that effectively convey its main findings. Its significance lies in challenging the dominant assumption of monotonic performance gains with scale and providing a new empirical framework for reasoning-specific scaling. The insights offered have potential to influence future research on model design and pretraining strategies aimed at enhancing reasoning rather than mere memorization.

**Weaknesses:**

The paper’s literature review appears incomplete because it omits several important recent seminal works on the science of language‐model reasoning, such as "When Scaling Meets LLM Finetuning: The Effect of Data, Model and Finetuning Method" (ICLR 2024), "Echo Chamber: RL Post-training Amplifies Behaviors Learned in Pretraining" (COLM 2025), "EvoLM: In Search of Lost Language Model Training Dynamics" (NeurIPS 2025), "Does Reinforcement Learning Really Incentivize Reasoning Capacity in LLMs Beyond the Base Model?" (NeurIPS 2025), all of which discuss how language model training affects reasoning. Without reference to these and other closely related studies, the work is less well‐situated in the broader research context.

The choice of a synthetic task setting somewhat raises concerns about the external validity of the findings. While the controlled environment is understandable, the paper does not include at least one experiment showing that performance on the synthetic reasoning task correlates with a meaningful real‐world reasoning benchmark (e.g. code, math), leaving open the question of how applicable the results are to practical large language model reasoning.

The experimental design lacks sufficient transparency and raises potential confounds: for example, the pretraining data size and how it scales (or not) with model size is not clearly specified, and it is unclear whether each model size was trained to full convergence. Without this information, the observed U-shaped relationship between model size and reasoning performance could be reflecting under‐training of larger models rather than an inherent reasoning‐versus‐memorization trade‐off. Moreover, a complete scaling law should ideally incorporate data amount (and compute) in addition to model size, which the paper does not address.

The interpretation of the drop in reasoning performance for larger models is potentially overly simplistic. The authors attribute this decline primarily to overparameterization and increased memorization, but they do not sufficiently explore alternative explanations (such as optimization difficulties, training regime mismatch, or capacity under‐utilization) nor provide deeper diagnostic analyses (like loss landscapes, generalization gaps, gradient behaviour) to support the claim. Also the theoretical construct of “graph search entropy” is indeed interesting but remains somewhat abstract and limited in interpretability. Although the paper presents an empirical linear relationship between this measure and optimal model size, it does not fully explain how the measure generalises to more complex or realistic graph‐structured data, and it does not explore the sensitivity of the empirical coefficient (~0.008 bits per parameter) to variations in task formulation or dataset design. This might raise question about how broadly applicable the proposed scaling law is beyond the constrained experimental setting.

Minor ones: while the figures and tables are clear, some of the axis labels and legneds are a bit small or lack units, which slightly hampers readability. Also, the paper occasionally uses jargon (for instance “search entropy per parameter”) without a brief intuitive definition in the main text, meaning readers not deeply familiar with information-theoretic measures may struggle. I would consider raise my scores if the above-mentioend points are addressed well.

**Questions:**

See weaknesses.

---

> ### Author Response · Authors · 2025-11-26
>
> We thank the reviewer for their feedback. Below is our response:
>
> 1. “The paper’s literature review appears incomplete”: Thank you for suggesting related articles. Due to the space limit, our related work section primarily focused on scaling law at the pretraining stage and implicit reasoning literatures which are the most relevant to our setting. The suggested articles are mostly focused on explicit reasoning with CoTs and post-training. We agree that a broader discussion of related work would better posit our findings. We have added a thorough discussion on the science of LM reasoning in the revision.
>
> 2. “ the paper does not include at least one experiment showing that performance on the synthetic reasoning task correlates with a meaningful real‐world reasoning benchmark (e.g. code, math)”: In additional to synthetic graphs, we show results on real-world knowledge graph, translated to natural language sentences in Figure 1. A similar but noisier U-shaped loss curved is observed. Here is an example of the pretraining data: American History X is featured under the drama film genre on Netflix. This is close to real-world pretraining data. This is in analogue to real world tasks like closed book multihop QA. Math and coding significantly diverge from our implicit reasoning setting as discussed in point 1 of general response.
>
> 3. “the pretraining data size and how it scales (or not) with model size is not clearly specified”: the training data size in Figure 1 is constant to be 300K. The training data size (N) in Figure 2 is shown in Table 3 in Appendix B. Figure 2 (b) and (f) have varying data sizes while others are constant.
>
> 4. “it is unclear whether each model size was trained to full convergence”: We confirm the larger models are trained to full convergence. In Figure 2 (a), the large models perform the best when the number of training steps is small. Larger number of training steps make it perform worse.
>
> 5. “a complete scaling law should ideally incorporate data amount (and compute) in addition to model size, which the paper does not address.”: graph search entropy is a more comprehensive description of data size. We did vary the training data size (Figure 2(b)(f)) and number of training steps (Figure 2(a)) along with other graph structures. We find the number of training steps (when large enough) does not affect the optimal model size.
>
> 6. “it does not fully explain how the measure generalizes to more complex or realistic graph‐structured data”: We show the scaling law accurately predicts the optimal model size of a real world KG (FB15K-237) in Figure 4.

---

### Author Response · Authors · 2025-11-26
**General response**

We thank all reviewers for their feedback. We want to clarify some important points upfront:

1. We are studying how language models generalize, with a specific task called IMPLICIT REASONING, as stated in the paper title as well as the abstract. We are NOT studying the reasoning tasks used in application oriented LLM papers, which usually means math word problems, coding and STEM QAs. These tasks require many specific types of reasoning and non-reasoning capabilities, including math computation, programming language generation, and natural language generation, which we do not study here in this paper. While the task name involves the word “reasoning”, it does not mean we are studying the application-oriented reasoning tasks. We have rewritten the abstract to make it clearer.
Note that, the task of implicit reasoning has appeared several times in other papers studying LLM reasoning/generalization, including [1-3], which confirms that this is a valid delegate task to study.

2. All models in the paper are pretrained FROM SCRATCH to rule out the effect of pre-learned reasoning capability from large pretraining corpus. The synthetic task is also constructed in a way to explicitly exclude the unwanted effect of natural language. E.g. some relations/entities may have similar names, and this will create unwanted correlations between them.

3. We are not trying to argue that smaller models ALWAYS perform better than larger models – with the correct training setting, it is obvious that large models would perform better than smaller models, and this has been shown in numerous studies. What we want to argue is that under appropriate training settings, small models can perform as well as large models (see figure 1(g)). Why do we want to study the setting where a smaller model performs well while a large model performs bad? First, we want to test the limit of reasoning capacity of language models: what is the smallest number of parameters that can reach the maximum reasoning performance? Second, larger models need more resources to train and serve. From an efficiency perspective, we want to always use the smallest model possible to reach the same performance.

4. There is no formal statement in the paper saying that the cause of U-shaped scaling law is “excessive memorization”. Rather, a similar sentence only appears ONCE in the abstract to intuitively introduce the phenomenon of U-shaped scaling law. What this “excessive memorization” refers to is that the training loss is able to monotonically decrease to a very small value when model size becomes larger as shown in figure 1 (c)(f)(i), while the testing loss shows an U-shape. Such an effect is also observed with synthetic KG, but due to the highly repetitive pattern, we didn’t include them in the paper. We have removed the word “memorization” from the abstract.


[1] Understanding Reasoning Ability of Language Models From the Perspective of Reasoning Paths Aggregation. Xinyi Wang, Alfonso Amayuelas, Kexun Zhang, Liangming Pan, Wenhu Chen, William Yang Wang. ICML 2024.

[2] Grokking of Implicit Reasoning in Transformers: A Mechanistic Journey to the Edge of Generalization. Boshi Wang, Xiang Yue, Yu Su, Huan Sun. NeurIPS 2024.

[3] How do Transformers Learn Implicit Reasoning? Jiaran Ye, Zijun Yao, Zhidian Huang, Liangming Pan, Jinxin Liu, Yushi Bai, Amy Xin, Liu Weichuan, Xiaoyin Che, Lei Hou, Juanzi Li. NeurIPS 2025.

---

### Meta-Review · Area_Chair_ERMM · 2026-01-10

**Summary:**

The decision to reject this paper is primarily based on significant concerns regarding the study's external validity and experimental robustness. Multiple reviewers questioned the generalizability of the findings, noting that the reliance on synthetic knowledge graph tasks and relatively small model architectures makes it difficult to extrapolate the observed "U-shaped" scaling law to modern, large-scale language models or real-world reasoning benchmarks (such as math or coding). Moreover, the experimental methodology faced scrutiny; specifically, some reviewers suggested that the performance degradation might be an artifact of the specific training regime, characterized by fixed steps and extensive data repetition (severe epoching), rather than a fundamental trade-off between reasoning and memorization. Consequently, the claim that overparameterization impairs implicit reasoning via "excessive memorization" was viewed as insufficiently substantiated by diagnostic metrics. In its present form, the submission does not currently meet the bar for acceptance at ICLR.

**Reviewer Concerns:**

The rebuttal successfully addressed several clarification points, including the precise definition of the "implicit reasoning" task, technical details regarding model convergence, and the inclusion of missing literature requested. The authors also clarified that the attribution of performance degradation to "excessive memorization" was intended as an intuitive description rather than a formal claim, and they agreed to amend the abstract accordingly. However, the most significant concerns regarding experimental validity and generalizability remain outstanding. The fundamental critique from the reviewers, that the reported "U-shaped" scaling law is likely an artifact of the specific training regime (characterized by "severe epoching" and fixed training steps) rather than a generalizable property, was not convincingly resolved. Moreover, the requests to establish external validity by testing on larger-scale models or diverse, real-world reasoning benchmarks were largely unmet, leaving substantial doubts about the applicability of these findings to modern LLM pretraining.

**Reviewer Scores:**

None of the reviewers responded to the authors' rebuttal. Reviewers CJHz and xsY7 likely would have maintained their marginally positive scores; the authors addressed their specific requests regarding missing literature and technical clarifications (such as the training objective), which may have solidified their support despite lingering external validity concerns. Reviewer nSeB would likely have maintained their strong rejection, as the rebuttal's defense of the "severe epoching" regime wasn't satisfactory. Similarly, Reviewers 1bdq and 7rMj would likely have kept their scores unchanged, as their substantial requests for experiments with larger models, real-world benchmarks, or stricter controls for data repetition were met with theoretical justifications for the existing study design rather than the requested empirical evidence.

---

### Decision · Program_Chairs · 2026-01-26

Reject